# Adaptive Internal Model Backstepping Control for a Class of Second-Order Electromagnetic Micromirror with Output Performance Constraints and Anomaly Control

**DOI:** 10.3390/mi15070925

**Published:** 2024-07-19

**Authors:** Huasen Gan, Yi Qin, Jinfeng Zhang, Cixing Lv, Zhonghua Chen, Yaohua Hu

**Affiliations:** 1School of Electrical Engineering and Intelligentization, Dongguan University of Technology, Dongguan 523000, China; 2College of Electronics and Information Engineering, Shenzhen University, Shenzhen 518000, China; 3Dongguan LongTTech Company Ltd., Dongguan 523000, China

**Keywords:** electromagnetic micromirror, output regulation, output performance constraint, anomaly control

## Abstract

This paper investigates the asymptotic tracking problem for a class of second-order electromagnetic micromirror model with output performance constraints and anomaly control, which is subject to model parameter uncertainties and external disturbances. Specifically, this paper formulates the trajectory tracking control problem of an electromagnetic micromirror as a closed-loop control trajectory tracking problem based on the general solution framework of output regulation. Moreover, the extended internal model is introduced to reformulate the closed-loop control problem into a state stabilization problem of the augmented system. Based on the augmented system, an internal model backstepping controller is proposed by integrating the barrier Lyapunov Functions (BLF) and the Nussbaum gain function with the backstepping structure.This controller not only satisfies the output performance constraints of the micromirror, but also maintains the control performance in anomalous control situations. The final performance simulation demonstrates the efficacy of the proposed controller.

## 1. Introduction

Micro-electro-mechanical systems (MEMS) is an interdisciplinary research field that centers on semiconductor technology and involves several disciplines such as mechanical engineering, microelectronics, materials science and control theory. In recent years, micro-optoelectromechanical systems (MOEMS), formed by the organic combination of MEMS technology and micro-optical technology, has become an essential research direction in the field of MEMS. In the research of MOEMS, MEMS micromirror, as the critical component, is extensively applied in novel optical system applications, showing great potential for applications in numerous fields such as optical switch [1], bio-medical display [2] and MEMS mirror-based Lidar [3].

MEMS micromirror is a sophisticated microelectromechanical system component consisting of micro-reflective mirrors, micromirror torsion bars, and drive actuators. With the advancement of MEMS technology, there are four main types of MEMS micromirrors on the market currently: electromagnetic-driven micromirror [4], electrostatic-driven micromirror [5], electrothermal-driven micromirror [6], and piezoelectric-driven micromirror [7]. Compared to the other three types of micromirrors, the electromagnetic-driven micromirror is suitable for micromirror mass production due to the performance advantages of high scanning frequency, low drive energy consumption and fast response speed. Consequently, MEMS electromagnetic micromirrors have a broad range of applications in the industrial sector, including but not limited to automotive LiDAR, micro-projection technology, augmented reality (AR), virtual reality (VR), and head-up display systems (HUD). The Yingtang Aurora Microtechnology has developed a two-dimensional electromagnetically driven MEMS micro-mirror product that has passed the reliability tests within the automotive industry, primarily aimed at medium and short-range vehicular LiDAR systems. The Zhongke Fusion Perception Intelligence Research Institute has launched its independently developed MEMS micromirror projection chip, which applies to 3D vision technology, micro-display imaging, automated guided vehicle (AGV), AR and VR applications.

At present, researchers in the field of MEMS have carried out a great deal of research work in enhancing the performance of micromirrors and have proposed a series of scientific and effective improvement schemes based on their respective research hypotheses. Active control primarily involves the design of micromirror hardware structures, such as the design of new drive structures and new micromirror structures [8,9,10]. Although these solutions can improve the system performance by refining the micromirror, the hardware solutions always require the utilization of more complex fabrication processes or more difficult-to-prepare materials. The manufacturing process has a low fault tolerance, the parameters are difficult to adjust, and the portability of the finished product is undesirable. Therefore, improving the performance of micromirror systems through hardware enhancements incurs significant costs and is accompanied by inherent limitations.

With the rapid development of modern optical applications, the tasks to be processed have become more complex, and the requirements for the tracking accuracy of optical micro-devices are becoming increasingly stringent. Under such circumstances, it is insufficient to rely solely on the optimisation of the micromirror hardware structure to improve the system performance. Since passive control needs to be considered, the focus is on maximizing the performance of the micromirror through the design of appropriate control algorithms. Passive control is an optimization at the algorithm level, which improves the response speed, accuracy and stability of the system by designing appropriate controllers without increasing the hardware cost. This method is effective in reducing cost while enhancing the dynamic performance of micromirrors and is highly portable. With the advancement of technology and the growing application requirements, research on micromirror control technology will continue to deepen to meet the needs of increasingly complex modern optical applications.

The rest of this paper is organized as follows: Section 2 presents the control research background and an overview of the research in this paper. Section 3 gives the second-order dynamic model of the micromirror system and defines the control objectives. Section 4 describes the design process of the adaptive internal mode backstepping controller and the analysis of asymptotic stability. Section 5 sets up simulation experiments to demonstrate the effectiveness of the proposed method, and the simulation results and analyses are presented. Section 6 proposes some concluding remarks.

## 2. Research Background

Traditional open-loop control is no longer suitable for application on MEMS high-precision devices due to its long stabilisation time and overshoot [11]. With the advancement of control theory in recent years, many novel control strategies have been proposed and applied to MEMS devices. Choi et al. implemented a model-based discrete-time Linear Quadratic Gaussian Regulator (LQGR) control scheme to achieve control over an optomechanical infrared imaging system [12]. Shi et al. proposed the zero vibration and zero derivative method (ZVD) based on input shaping techniques to address the control issues of thermoelectric micromirrors [13]. Zhou et al. developed a sliding mode controller based on the exponential convergence law for the control of an electromagnetically driven micromirror, which improved the control response speed of the micromirror [14]. Weinberger et al. presented a flatness-based approach to solve the problem of controlling electrostatically driven micromirrors [15]. Dong et al. designed a dual-axis $H_{\infty }$ robust controller to implement the control of the portable Fourier Transform spectral Acquisition Systems primarily composed of MEMS micromirrors [16]. Qin et al. utilized a PID sliding mode control algorithm for electromagnetic micromirror control, which resulted in superior transient response and tracking accuracy compared to traditional PID control [17]. The aforementioned solutions can address their respective control issues, but they also have some inevitable shortcomings. The Linear Quadratic Regulator (LQR) control strategy is computationally demanding and requires that the known characteristics of the system must not be altered, otherwise it can lead to errors in the control algorithm. Control methods based on input shaping typically rely on accurate modeling of system behavior and consequently are sensitive to variations of the system parameters. The performance of the input shaping control and the $H_{\infty }$ control approach may be negatively impacted by system parameter variations or uncertainties, which can result in control accuracy degradation or stability problems. Although sliding mode control can resolve the aforementioned system problems due to variations in system parameters, the law-switching property of sliding mode control causes the system to chatter. Therefore, sliding mode control is generally incorporated in conjunction with other control techniques to mitigate the chattering phenomenon [17,18]. Nevertheless the complex design of sliding mode surfaces and switching conditions introduces the problem of electromagnetic compatibility noise.

In addition, most of the existing control research for electromagnetic micromirrors are based on error information. However, micromirrors have fast dynamic characteristics, which means that sensor noise is introduced when the error differential information is actually acquired. This uncertain noise is further amplified during the differentiation process, causing the performance of controllers designed based on the error differential information from simulation experiments to fall short of the actual control performance. Therefore, it is necessary to reduce the dependence on the higher-order differential information of the output error. Tan et al. addressed the control of electromagnetic micromirrors based on the idea of output regulation, using a classical internal model to eliminate the need for higher-order differential information of the output error, while solving the problem based on angular velocity information [19]. Subsequently, Sun et al. targeted the problem of the angular velocity information of the electromagnetic micromirror being difficult to measure accurately, defined the micromirror tracking as an output regulation problem, and applied the extended internal model to formulate the output regulation problem as a state stability problem for solution [20]. Compared with the work in [19], the control algorithm proposed in [20] has a significant advantage in that it can realize asymptotic tracking of the micromirror by relying solely on angular information. This approach simplifies the design of the control system and reduces the use of sensors, thereby making the control algorithm easier to be integrated and applied to the control of the actual micromirror and reducing the system cost.

The issue to be solved in this paper is the asymptotic tracking control problem for the electromagnetic micromirror with output performance constraints and unknown control directions. On one hand, there are physical structure constraints of the micromirror itself. Before reaching a stable scan, excessive deflection angles can cause the micro-reflector to collide with the planar microcoil, thereby damaging the micromirror and shortening its lifetime [21]. The barrier Lyapunov function [22] is an effective solution for the constrained question. In recent years, the BLF has been used as a control theory tool in control areas such as robot arms to ensure that the robotic arm accurately tracks the predetermined path, thus improving the safety and efficiency of the operation [23]. There is a significant difference in the mathematical properties of the BLF and the Quadratic Lyapunov Function (QLF). The BLF is designed to grow rapidly to infinity as the state variable approaches a predetermined boundary or parameter value [24]. This design of the BLF allows it to provide stronger control of the state variable as the state approaches the boundary, ensuring system stability and safety. In addition, the BLF demonstrates faster convergence while satisfying the performance constraints, and the initial conditions required to achieve the same performance constraints are more relaxed than those of the QLF.

On the other hand, the actual micromirror may encounter the question of uncertainty in the direction of control after prolonged use [25], which is referred to in this paper as anomalous control of the micromirror. This is caused by an anomaly in the Field Programmable Gate Array (FPGA) or Voltage Controlled Current Amplifier (VCCA) circuit that causes the control system to need to adapt to this unknown change in control direction. When a control abnormality occurs, the VCCA circuit supplies the wrong current to the planar drive coil, which generates the opposite control force, thereby affecting the control performance of the micromirror. The Nussbaum gain technique is a standard method for dealing with the problem of unknown control directions. The Nussbaum gain technique is a technique for dealing with unknown or changing control directions in adaptive control. Specifically, the controller degrades the system performance when the control direction of the controller is opposite to the desired direction of the system. The controller provides higher gain when the control direction is corrected to the proper direction, allowing the system to converge to the desired state more quickly [26]. In recent years, the Nussbaum function has been successfully combined with other control techniques to solve control issues in the presence of unknown control directions. Tan et al. employed a combination of the Nussbaum function and backstepping to successfully solve the adaptive control of a micromirror system [25]. Luo et al. proposed a solution to the problem of adaptive control of an arch MEMS resonator by combining the Nussbaum function, neural network and extended state observer [27].

Inspired by the aforementioned work, this paper addresses the asymptotic tracking control problem of an electromagnetic micromirror that considers output performance constraints and anomaly control. In this paper, the reference signal is firstly represented as an external system signal based on the output regulation theory [28], which is combined with the micromirror system to construct a closed-loop control system. Through state transformation, the closed-loop control problem is transformed into a lower triangular form system, preparing the ground for the subsequent application of an extended internal model. Subsequently, the extended internal model of the lower triangular system is constructed. Combining the extended internal model with the lower triangular system, the system control problem is converted into the state stability problem of the augmented system through further state transformation. On this basis, BLF and Nussbaum gain techniques are comprehensively applied to solve the problems of the electromagnetic micromirror system in terms of output performance constraints and anomaly control.

The major contributions of this paper can be summarized as follows: (i) In order to reduce the cost of the system and to minimise the reliance on observers and sensors, in contrast to the controller established in [19] with the classical internal model, this paper draws on the extended internal model introduced in [20]. We propose an adaptive backstepping output feedback control scheme based on an extended internal model to achieve stable control of the micromirror system. (ii) The designed controller takes into account system parameter uncertainties and external disturbances while dealing with input saturation. (iii) Compared to the work in [20,25], the controller design proposed in this paper comprehensively considers the impact of output performance constraints and anomaly control on the micromirror system. The advantage of this design is that it can ensure the output performance of the micromirror system even if an anomaly occurs during the control process, and protect the micromirror system from damage by avoiding collision between the micromirror and the planar drive coil due to the excessive deflection angle.

## 3. Micromirror Dynamics and Problem Definition

The entire structure of the micromirror is depicted in Figure 1, with the main components including the micro reflector, torsion bar, drive coil, and base. The torsion bar is connected to the micro reflector and fixed to the base to provide stable support. The drive coil is located below the micro reflector, working in conjunction with the torsion bar and the micro reflector. The structure of the VCCA circuit for the electromagnetic micromirror is presented in Figure 2. Figure 3 and Figure 4 display the overhead and lefthand views of the micromirror, respectively, with $M_{1}M_{1}^{\prime }$, $M_{1}^{\prime }M_{3}^{\prime }$, $M_{3}M_{3}^{\prime }$ and $M_{1}M_{3}$ representing the magnetic films. The components of the micromirror system platform are illustrated in Figure 5.

The composite second-order dynamic model that describes the dynamic behavior of the micromirror system is provided by [29]:(1)$$I_{em}\ddot{\Theta }+D_{em}\dot{\Theta }+S_{em}\Theta =T_{field}\left(\Theta \right)$$
where Θ represents the deflection angle, $\dot{\Theta }$ denotes the angular velocity and $\ddot{\Theta }$ signifies the angular acceleration. $I_{em}$ is the rotational inertia. $D_{em}$ is the damping coefficient. $S_{em}$ is the spring torsion bar coefficient. $T_{field}$ is the electromagnetic resultant moment of force.

The electromagnetic torque $T_{field}$ is generated by the drive current *i*. Qin et al. utilized the finite element analysis (FEA) method to model a finite element model of the electromagnetically driven micromirror, and concluded that there is an almost linear relationship for the electromagnetic torque $T_{field}$ and the driving current *i* [17]. The drive current is thus used as the control input for the model (Equation 1). In addition, the electromagnetic micromirror operates with a maximum drive current of 1A [30].

Considering the above conditions, we define $z_{1}=\Theta$, $z_{2}=\dot{\Theta }$ and sat(u)=i, where $sat\left(u\right)\leq |u_{m}|$ and $u_{m}=1$. Thus, the dynamics of the electromagnetic micromirror is modelled as
(2)$$\begin{array}{cc} {\dot{z}}_{1} & =z_{2}\\ {\dot{z}}_{2} & =-\frac{S_{em}}{I_{em}}z_{1}-\frac{D_{em}}{I_{em}}z_{2}+\frac{b}{I_{em}}sat\left(u\right)+d_{e} \end{array}$$
where $\frac{b}{I_{em}}$ denotes the control input gain. b≠0 and its sign is indeterminate. $d_{e}$ represents the external disturbance. Considering the changes in system parameters, $\Delta =(D_{em},S_{em},I_{em},b)$ is set as the actual values of parameter vectors. $\hat{\Delta }=\left({\hat{D}}_{em},{\hat{S}}_{em},{\hat{I}}_{em},\hat{b}\right)$ is set as the nominal values of parameter vectors. Then $\Delta =\hat{\Delta }+\lambda$, where $\lambda \in \Lambda \in {ℜ}^{4}$ denotes the change value of the model parameter vector.

To facilitate the subsequent design of the extended internal model, the saturation function sat(u), a nonlinear term, needs to be dealt with. We define sat(u) to be $g_{u}\left(\xi \right)=u_{m}$**tanh**$\left(\frac{\xi }{u_{m}}\right)$, where $u_{m}=1$. Based on the general properties of the micromirror system, we choose $g_{u}\left(\xi \right)$ to be linearised near the origin. Combining the aforementioned conditions, we get $g_{u}\left(\xi \right)=\xi$. The uncertainty term $d_{c}=\mathrm{sat}\left(u\right)-g_{u}\left(\xi \right)$ is then defined, and $d_{c}$ is bounded. The micromirror system is therefore depicted as follows:(3)$$\begin{array}{cc} {\dot{z}}_{1} & =z_{2}\\ {\dot{z}}_{2} & =-\frac{S_{em}}{I_{em}}z_{1}-\frac{D_{em}}{I_{em}}z_{2}+\frac{b}{I_{em}}\xi +D \end{array}$$
where $D=d_{e}+\frac{b}{I_{em}}d_{c}$ and *D* is bounded. The control law design task now transforms into determining the auxiliary control variable ξ.

The ability of micromirrors to produce grating patterns for high-quality imaging is predicated on ensuring that the micromirror can asymptotically track a given reference trajectory, e.g., a sinusoidal signal trajectory. In the framework of output regulation solving based on internal model control, the reference trajectory signal is generally generated by an external system. Therefore, we can define the reference signal of the micromirror in the following form:(4)$$\dot{h}=Sh$$
where $h=\left[\begin{array}{c} h_{1}\\ h_{2} \end{array}\right]$ and $S=\left[\begin{array}{cc} 0 & \sigma \\ -\sigma & 0 \end{array}\right]$. The corresponding reference signal is $F_{ref}\left(t\right)=h_{1}\left(t\right)$, where amplitude $A_{em}=\sqrt{h_{1}^{2}\left(0\right)+h_{2}^{2}\left(0\right)}$ and phase $\varphi_{em}$ = **arctan**$\frac{h_{1}\left(0\right)}{h_{2}\left(0\right)}$. All eigenvalues of *S* are unique and the real part is zero.

The system (Equation 4) is integrated with the system (Equation 3) to form a closed-loop control system. The micromirror progressive tracking control problem can be formulated as follows:(5)$$\begin{array}{cc} {\dot{z}}_{1} & =z_{2}\\ {\dot{z}}_{2} & =-\frac{S_{em}}{I_{em}}z_{1}-\frac{D_{em}}{I_{em}}z_{2}+\frac{b}{I_{em}}\xi +D\\ \dot{h} & =Sh\\ e & =z_{1}-h_{1} \end{array}$$
where *e* represents the angular error.

The subsequent parts of this investigation, including the design of the controller and the demonstration of stability, are based on system (Equation 5). The goal of this paper is to design a suitable control strategy that ensures that a micromirror system with output performance limitations and unknown control directions is capable of achieving asymptotic tracking of the reference signal $F_{ref}\left(t\right)$. Specifically, we want the system to satisfy the following conditions: $\lim_{t\to \infty }e\left(t\right)=0$, where $\left|e(t)\right|<k_{m}\left(t\right)$. $k_{m}\left(t\right)$ is the error threshold to be set.

## 4. Control Strategy Development

The design of the controller is separated into two sections. The first section involves the construction of an extended internal model control structure capable of generating auxiliary control signals. This extended internal model is combined with system (Equation 5) and further state transformations are performed to obtain a new extended system. In this way, we transform the closed-loop asymptotic control problem for micromirrors into research on the state stability of the new extended system.

The second section is based on the augmented system and adopts the backstepping method as the basic structure of the control strategy. Within this structure, the BLF and the Nussbaum gain function are integrated to address the problems of state constraints and anomaly control. With this approach, we are able to design a control strategy that satisfies the performance requirements and adapts to the system uncertainty.

### 4.1. Extended Internal Model and System Transformation

Following the idea of constructing an extended internal model in [20,31], we are required to transform the system into a lower triangular form. To achieve this goal, we introduce a specific filter (Equation 6) and state transition (Equation 7), which facilitate the subsequent control design and analysis.
(6)$$\dot{m}=-m+\xi$$
(7)$$z=z_{2}-K\left(\lambda \right)z_{1}-L\left(\lambda \right)m$$
where $K\left(\lambda \right)=-\frac{D_{em}}{I_{em}}+1$ and $L\left(\lambda \right)=\frac{b}{I_{em}}$. Substituting (Equation 6) and (Equation 7) into system (Equation 5), we obtain the following lower triangular form of the system.
(8)$$\begin{array}{cc} \dot{z} & =-z+J\left(\lambda \right)z_{1}+D\left(h,\lambda \right)\\ {\dot{z}}_{1} & =z+K\left(\lambda \right)z_{1}+L\left(\lambda \right)m\\ \dot{m} & =-m+\xi \\ \dot{h} & =Sh\\ e & =z_{1}-h_{1} \end{array}$$
where $J\left(\lambda \right)=-(K\left(\lambda \right)+\frac{S_{em}}{I_{em}})$. D(h,λ) is sufficiently smooth and D(0,λ)=0.

To successfully construct an extended internal model for system (Equation 5), it is imperative to establish a set of foundational assumptions initially.

**Assumption** **1.**
*There exists a sufficiently smooth function *
*
**z**
*
*(h, λ) that meets the initial condition *
*
**z**
*
*(0, λ) = 0 for $h\in {ℜ}^{2}$, $\lambda \in {ℜ}^{4}$ and σ∈ℜ. The function *
*
**z**
*
*(h, λ) satisfies the following differential equation:*

(9)
$$\frac{\partial z(h,\lambda )}{\partial h}Sh=-z\left(h,\lambda \right)+J\left(Z_{1}\left(h,\lambda \right),h,\lambda \right)+D\left(h,\lambda \right)$$



**Assumption** **2.**
*The function ϱ(h,λ) is a polynomial function in the variable h, with the polynomial coefficients varying as a function of the uncertain parameter λ.*


According to the theory of output regulation [28], the solvability of the output regulation problem is contingent upon the solvability of the regulator equations associated with system (Equation 8). The regulator equation corresponding to system (Equation 8) is as follows:(10)$$\begin{array}{cc} \frac{\partial z(h,\lambda )}{\partial h}Sh & =-z\left(h,\lambda \right)+J\left(\lambda \right)Z_{1}\left(h,\lambda \right)+D\left(h,\lambda \right)\\ \frac{\partial Z_{1}\left(h,\lambda \right)}{\partial h}Sh & =-z\left(h,\lambda \right)+K\left(\lambda \right)Z_{1}\left(h,\lambda \right)+L\left(\lambda \right)M\left(h,\lambda \right)\\ \frac{\partial M(h,\lambda )}{\partial h}Sh & =-M(h,\lambda )+\varrho (h,\lambda )\\ 0 & =Z_{1}\left(h,\lambda \right)-h_{1} \end{array}$$
where z(h,λ) and $Z_{1}\left(h,\lambda \right)$ represent the zero-error steady state information of the micromirror system. M(h,λ) denotes the control input required to achieve the desired angle tracking of the micro-mirror, thus attaining the zero-error steady-state information. Assumption 1 ensures the solvability of the regulator Equation (Equation 8).

Nevertheless, the presence of uncertain parameters *b* and λ renders the direct application of z(h,λ), $Z_{1}\left(h,\lambda \right)$ and M(h,λ) to the micro-mirror system impractical. These functions encapsulate the steady-state information and the control input required to achieve zero-error performance under ideal conditions. Consequently, the introduction of a dynamic compensator is required to estimate the control input M(h,λ). Assumption 2 ensures the feasibility of constructing a dynamic compensator capable of accommodating the uncertainty of the micromirror system. This dynamic compensator is the extended internal model to be designed.

Following the approach to establishing an internal model as outlined in [20,32], we can construct the extended internal model through the following procedure.

Defining z(h,λ) = $C_{1}\left(\lambda \right)h$, and $C_{1}$ satisfies the following equation:(11)$$C_{1}\left(\lambda \right)S=-C_{1}\left(\lambda \right)+J$$
where $C_{1}$ = [$c_{11}$$c_{12}$] and *J* = [J(λ) 0]. Expanding the above equation yields $c_{11}$ and $c_{12}$ as
(12)$$\begin{array}{cc} c_{11} & =\frac{J(\lambda )}{1+\sigma^{2}}\\ c_{12} & =-\frac{\sigma J(\lambda )}{1+\sigma^{2}} \end{array}$$

With the second equation of the regulator Equation (Equation 10), we can get
(13)$$M\left(h,\lambda \right)=L^{-1}\left(\lambda \right)\left(\frac{\partial Z_{1}\left(h,\lambda \right)}{\partial h}Sh-z\left(h,\lambda \right)-K\left(\lambda \right)Z_{1}\left(h\right)\right)$$

Then define M(h,λ) = $C_{2}\left(\lambda \right)h$, where $C_{2}\left(\lambda \right)$ = [$c_{21}$$c_{22}$]. Substituting M(h,λ) into the above Equation (Equation 13) yields
(14)$$\begin{array}{cc} c_{21} & =-\frac{1}{L(\lambda )}\left(c_{11}+K\left(\lambda \right)\right)\\ c_{22} & =\frac{1}{L(\lambda )}\left(\sigma -c_{12}\right) \end{array}$$

As posited by [28], under the premise of Assumption 2, there exists a polynomial with real coefficients and an integer value denoted by *ℏ* such that
(15)$$P\left(\tau_{c}\right)=\tau_{c}^{\hbar }-a_{1}-a_{2}\tau_{c}-...-a_{\hbar }\tau_{c}^{\hbar -1}$$

Meanwhile, M(h,λ) fulfils
(16)$$\frac{d^{\hbar }M\left(h,\lambda \right)}{dt^{\hbar }}=a_{1}M\left(h,\lambda \right)+a_{2}\frac{dM(h,\lambda )}{dt}+...+a_{\hbar }\frac{d^{\hbar -1}M\left(h,\lambda \right)}{dt^{\hbar -1}}$$
for all h(t) associated with the exosystem (Equation 4) and ∀λ∈Λ.

Subsequently, define the matrix $\Psi =\left[\begin{array}{cc} 0 & 1\\ a_{1} & a_{2} \end{array}\right]$ and the vector *Q* = [1 0]. For any given controllable pair (Φ, *E*), where Φ is a Hurwitz matrix and *E* is a column vector, there exists a non-singular matrix Ω such that the Sylvester equation [33] is satisfied. Optimization of control system performance can be achieved by selecting different pairings (Φ, *E*). The Sylvester equation is given as follows:(17)ΩΨ−ΦΩ=EQ

**Remark** **1.***In control theory, a Hurwitz matrix is a square matrix with real coefficients and all its eigenvalues are in the left half of the open region of the complex plane.* Φ *is a Hurwitz matrix which satisfies the following conditions: It is a square matrix and the real part of all its eigenvalues are negative. The matrix* Φ *is an adjustable control parameter and the setting of the parameter does not depend on other variables. For example,* Φ *can be set to $\left[\begin{array}{cc} -a_{\phi 1} & 0\\ 0 & -a_{\phi 2} \end{array}\right]$, where $a_{\phi 1}$ and $a_{\phi 2}$ are positive numbers.*

Finally, let N(h,λ) = Ω[M(h,λ), $\dot{M}{\left(h,\lambda \right)]}^{T}$. Following the solution approach for the steady-state generator presented in [28], the following derivations are established:(18)$$\begin{array}{c} \frac{dN(h,\lambda )}{dt}=\Omega \Psi \Omega^{-1}N\left(h,\lambda \right)\\ M\left(h,\lambda \right)=Q\Omega^{-1}N\left(h,\lambda \right) \end{array}$$

Following the formulation presented in Equation (Equation 18), the extended internal model of the system (Equation 8) can be formulated as:(19)$$\dot{\eta }=\Phi \eta +Em$$

The aforementioned extended internal model (Equation 19) is capable of providing an asymptotic estimate of N(r,λ). By integrating this model with the system (Equation 8), a generalized system is formed. This integration is further facilitated through the introduction of the following state transformation:(20)$$\begin{array}{cc} \bar{z} & =z-z(r,\lambda )\\ e & =z_{1}-h_{1}\\ \bar{m} & =m-\beta_{1}\left(\eta \right)\\ \bar{\eta } & =\eta -N-El^{-1}e\\ \bar{\xi } & =\xi -\beta_{2}\left(\eta \right) \end{array}$$
where N=N(h,λ), l=L(λ), $\beta_{1}\left(\eta \right)=Q\Omega^{-1}N$ and $\beta_{2}\left(\eta \right)=\beta_{1}\left(\eta \right)+\frac{\partial \beta_{1}\left(\eta \right)}{\partial \eta }\dot{\eta }$, which leads to the formulation of the augmented system (Equation 21).
(21)$$\begin{array}{cc} \dot{\bar{z}} & =-\bar{z}+J\left(\lambda \right)e\\ \dot{\bar{\eta }} & =\Phi \bar{\eta }+\Phi El^{-1}e-l^{-1}E\left(\bar{z}+K\left(\lambda \right)e\right)\\ \dot{e} & =\bar{z}+K\left(\lambda \right)e+Q\Omega^{-1}\left(Ee+l\bar{\eta }\right)+l\bar{m}\\ \dot{\bar{m}} & =-\left(1+Q\Omega^{-1}E\right)\bar{m}+\bar{\xi } \end{array}$$

### 4.2. Stability Solving for Augmented System

Building upon the research conducted in the preceding sections, the trajectory tracking problem of the electromagnetic micro-mirror has been effectively reformulated as an issue of state stability within an augmented system (Equation 21). To address the challenges of state constraints and anomaly control, this section will further explore the integration of the BLF and the Nussbaum gain techniques within the backstepping control structure. The synergistic application of these two methodologies offers an efficacious solution to the trajectory tracking problem of the electromagnetic micromirror, particularly when the system is subject to output performance constraints and anomaly control.

Before proceeding with the controller design, there are some necessary definitions and lemmas that need to be introduced in order to establish the theoretical foundations and provide support for the subsequent derivations.

**Definition** **1**([34]). *The function *ℵ*(ς) is characterized as a smooth R-type Nussbaum function (otherwise known as a Ryan-type Nussbaum Gain Function). The R-type Nussbaum function is a more conditional Nussbaum function with the properties outlined below:*
(22)$$\begin{array}{cc} \lim_{r\to \infty }\sup\frac{1}{r}\int_{0}^{r}ℵ\left(\varsigma \right)d\varsigma & =\infty \\ \lim_{r\to \infty }\inf\frac{1}{r}\int_{0}^{r}ℵ\left(\varsigma \right)d\varsigma & =-\infty \end{array}$$

**Remark** **2.**
*Based on the above properties of R-type Nussbaum functions, we can derive some examples of R-type Nussbaum functions that satisfy these properties. For example, $\varsigma^{2}\cos\left(\varsigma \right)$ and $\varsigma^{2}\sin\left(\varsigma \right)$, etc. The R-type Nussbaum function can be used to solve several system calibration problems with uncertain control coefficients, which combine with the following lemma 1 to guarantee the boundedness of the Lyapunov-like energy function.*


**Definition** **2**([35]). *Let ℵ(ς) be defined as an R-type Nussbaum function. For any bounded function $f_{b}\left(\cdot \right)$, the function $ℵ\left(\varsigma \right)+f_{b}\left(\cdot \right)$ belongs to the R-type Nussbaum function.*

**Definition** **3**([20]). *Given that J(λ)e and K(λ)e are continuous real-valued functions for λ∈Λ, it follows that there exist constants $q_{1}$ and $q_{2}$ such that the following inequalities hold:*
(23)$$\begin{array}{c} {\left|J\left(\lambda \right)e\right|}^{2}\leq q_{1}e^{2},{\left|K\left(\lambda \right)e\right|}^{2}\leq q_{2}e^{2} \end{array}$$

**Lemma** **1**([36]). *Let V(·) and ξ(·) be smooth functions defined on the interval [0,T), and let* ℵ*(ξ) be a Nussbaum-type function. If the following inequality is satisfied:*
(24)$$0\leq V\left(t\right)\leq \int_{0}^{t}\left(lℵ\left(\xi \left(\cdot \right)\right)+\kappa_{1}\right)\dot{\xi }\left(\cdot \right)dt+\kappa_{2},t\in \left[0,T\right)$$
*where $\kappa_{1}$ is a positive constant, $\kappa_{2}$ is a constant. Then it can be concluded that V(·), ξ(·) and $\int_{0}^{t}\left(lℵ\left(\xi \left(\cdot \right)\right)+\kappa_{1}\right)\dot{\xi }\left(\cdot \right)$ are all bounded on the interval [0,T).*

**Lemma** **2**([22]). *Let $\varepsilon_{u}$ and $\varepsilon_{l}$ be positive constants, and define the open interval* Π *as ${ℜ}^{l}\times Z\subset {ℜ}^{l+1}$ and G as the set {$g\in ℜ:-\varepsilon_{l}<g<\varepsilon_{u}$ } ⊂ℜ. Consider the following system*
(25)$$\dot{\eta }=f\left(t,\eta \right)$$
*where $\eta ={\left[\lambda ,g\right]}^{T}\in \Pi$ and the function $f:{ℜ}_{+}\times \Pi \to {ℜ}^{l+1}$ is locally Lipschitz in η and piecewise continuous in t. Assume the existence of two differentiable functions $V_{1}:G\to {ℜ}_{+}$ and $U:{ℜ}^{l}\to {ℜ}_{+}$ that satisfy the following conditions: 1) $V_{1}\left(g\right)\to \infty$, for $g\to -\varepsilon_{l}$ or $g\to \varepsilon_{u}$; 2) $\phi_{1}$ and $\phi_{2}$ are class* **K_∞_**
*functions, and $\phi_{1}\left(∥\lambda ∥\right)\leq U\left(\lambda \right)\leq \phi_{2}\left(∥\lambda ∥\right)$. Define the function $V\left(\eta \right)=U\left(\lambda \right)+V_{1}\left(g\right)$ with the initial condition g(0)∈G. For g∈G, if $\dot{V}=\frac{\partial V}{\partial \eta }f\leq 0$, then g(t)∈G with ∀t∈[0,∞).*

The BLF chosen is as follows:(26)$$V_{BLF}=\frac{1}{2}\ln\frac{k_{m}^{2}\exp\left(x^{2}\right)}{k_{m}^{2}-x^{2}}$$
where ln(·) represents the natural logarithm and exp(·) denotes the exponential function with the natural constant **e** as the base. $k_{m}$ is the state threshold to be set and *x* is the state of the system.

**Remark** **3.***The BLF presented in* (Equation 26) *offers distinct advantages over conventional BLFs, such as $\frac{1}{2}\ln\frac{k_{m}^{2}}{k_{m}^{2}-x^{2}}$, $\frac{k_{m}^{2}}{2\pi }\tan\left(\frac{\pi }{2}{\left(\frac{x}{k_{m}}\right)}^{2}\right)$ and $\exp(\frac{x^{2}}{k_{m}^{2}-x^{2}}-1)$. Specifically, the BLF in* (Equation 26) *exhibits two principal advantages: Firstly, this BLF is able to effectively handle both cases with and without output constraints, which provides flexibility in controller design. Secondly, the number of Gaussian terms to be handled during the derivation of the controller is reduced, i.e., it ensures the performance of the system and at the same time lowers the complexity of the controller design, which is why this BLF is chosen to be applied to the controller design in this paper.*

**Remark** **4.**
*The $k_{m}$ can be a variable function as well as a constant function, which needs to be considered according to the actual situation. After weighing the desired output performance and the control input, $k_{m}$ is set as a constant in this paper.*


The Nussbaum-type gain function selected for this paper is
(27)$$ℵ\left(\varsigma \right)=\varsigma^{2}\cos\left(\varsigma \right)$$

The zero dynamics depends on the subsystem ($\bar{z}$, $\bar{\eta }$). Moreover, for ∀λ, when *e* is the input, the zero dynamics is ISS (input-to-state stable, within the meaning of [37]). Along the lines of [20], define
(28)$$U={\bar{k}}_{1}\;{\bar{z}}^{T}\bar{z}+{\bar{k}}_{2}\;{\bar{\eta }}^{T}\Gamma \bar{\eta }$$
where ${\bar{k}}_{1}$ and ${\bar{k}}_{2}$ are positive constant and matrix Γ satisfies $\Gamma \Phi +\Phi^{T}\Gamma =-I_{2}$, where Φ is a Hurwitz matrix, $I_{2}$ is a unit matrix. Therefore, $\dot{U}$ along the subsystem of system (Equation 21) satisfies
(29)$$\begin{array}{cc} \dot{U}\leq & -\left({\bar{k}}_{1}\left(1-\delta_{1}\right)-{\bar{k}}_{2}\delta_{2}^{-1}\right)∥\bar{z}{∥}^{2}-{\bar{k}}_{2}(1-\delta_{2}∥\Gamma \Phi El^{-1}{∥}^{2}-2\delta_{2}∥\Gamma El^{-1}{∥}^{2})∥\bar{\eta }{∥}^{2}\\ \; & +\left({\bar{k}}_{1}\delta_{1}^{-1}q_{1}+{\bar{k}}_{2}\delta_{2}^{-1}\left(q_{2}+1\right)\right)e^{2}\\ \leq & -k_{1}∥\bar{z}{∥}^{2}-k_{2}{∥\bar{\eta }∥}^{2}+ge^{2} \end{array}$$
where $k_{1}={\bar{k}}_{1}\left(1-\delta_{1}\right)-{\bar{k}}_{2}\delta_{2}^{-1}$, $k_{2}={\bar{k}}_{2}(1-\delta_{2}∥\Gamma \Phi El^{-1}{∥}^{2}-2\delta_{2}∥\Gamma El^{-1}{∥}^{2})$ and $g>{\bar{k}}_{1}\delta_{1}^{-1}q_{1}+{\bar{k}}_{2}\delta_{2}^{-1}\left(q_{2}+1\right)$.

Based on the above definition, let $x_{1}=e$, $x_{2}=\bar{m}$. Then the output feedback controller of the micromirror is developed as
(30)$$\begin{array}{cc} \bar{\xi } & =\left(1+Q\Omega^{-1}E\right)x_{2}-{\bar{x}}_{2}+\hat{l}\left(\frac{\partial \alpha }{\partial x_{1}}x_{2}-x_{1}\right)-{\left(\frac{\partial \alpha }{\partial x_{1}}\right)}^{2}{\bar{x}}_{2}+\frac{\partial \alpha }{\partial \nu }\dot{\nu }+\frac{\partial \alpha }{\partial k_{m}}\dot{k_{m}}\\ \alpha & =\frac{k_{m}^{2}-x_{1}^{2}}{k_{m}^{2}-x_{1}^{2}+1}ℵ\left(\nu \right)\rho \left(x_{1}\right)x_{1}\\ \dot{\hat{l}} & =-(\frac{\partial \alpha }{\partial x_{1}}{\bar{x}}_{2}x_{2}-{\bar{x}}_{2}x_{1})\\ \dot{\nu } & =x_{1}^{2}\rho \left(x_{1}\right) \end{array}$$
where $\rho \left(x_{1}\right)=1+\frac{1}{2}\sin\left(k_{m}^{2}-x_{1}^{2}\right)$ and ${\bar{x}}_{2}=x_{2}-\alpha$. The designed controller can effectively solve the state stabilisation problem of system (Equation 21), while equivalently solving the closed-loop asymptotic tracking control problem of system (Equation 5) as well as the control problem of the electromagnetic micromirror system (Equation 1).

In this paper, we prove the stability of the system by defining the Lyapunov functions $V_{1}$ and $V_{2}$, combining Lemmas 1 and 2. The argumentation process is as follows:

**Proof.** Let $V_{1}=U+\frac{1}{2}\ln\frac{k_{m}^{2}\exp\left(x_{1}^{2}\right)}{k_{m}^{2}-x_{1}^{2}}$, then we have
(31)$$\begin{array}{cc} \dot{V_{1}}= & \;\dot{U}+\frac{x_{1}\left(k_{c}^{2}-x_{1}^{2}+1\right)}{k_{c}^{2}-x_{1}^{2}}(\bar{z}+K\left(\lambda \right)x_{1}+Q\Omega^{-1}Ex_{1}\\ \; & +lQ\Omega^{-1}\bar{\eta })-\frac{\dot{k_{m}}}{k_{m}}\frac{x_{1}^{2}}{k_{m}^{2}-x_{1}^{2}}+\frac{x_{1}\left(k_{m}^{2}-x_{1}^{2}+1\right)}{k_{m}^{2}-x_{1}^{2}}l({\bar{x}}_{2}\\ \; & +\frac{k_{c}^{2}-x_{1}^{2}}{k_{c}^{2}-x_{1}^{2}+1}ℵ\left(\nu \right)\rho \left(x_{1}\right)x_{1}) \end{array}$$Let $V_{2}=V_{1}+{\bar{x}}_{2}^{2}+{\left(\hat{l}-l\right)}^{2}$, then we get
(32)$$\begin{array}{cc} \dot{V_{2}}= & \;\dot{V_{1}}+2{\bar{x}}_{2}(-\left(1+Q\Omega^{-1}E\right)x_{2}+\bar{\xi }-\frac{\partial \alpha }{\partial \nu }\dot{\nu }-\frac{\partial \alpha }{\partial k_{m}}\dot{k_{m}}\\ \; & -\frac{\partial \alpha }{\partial x_{1}}\left(\bar{z}+K\left(\lambda \right)x_{1}+Q\Omega^{-1}Ex_{1}+lQ\Omega^{-1}\bar{\eta }+lx_{2}\right))\\ \; & +2\left(\hat{l}-l\right)\dot{\hat{l}} \end{array}$$By utilizing (Equation 30) and (Equation 31), combined with Young’s inequality, $\dot{V_{2}}$ is deduced as
(33)$$\begin{array}{cc} \dot{V_{2}}= & \;\dot{V_{1}}-2{\bar{x}}_{2}\frac{\partial \alpha }{\partial x_{1}}\left(\bar{z}+K\left(\lambda \right)x_{1}+Q\Omega^{-1}Ex_{1}+lQ\Omega^{-1}\bar{\eta }\right)\\ \; & -2l{\bar{x}}_{2}x_{1}-2{\bar{x}}_{2}\left({\bar{x}}_{2}+{\left(\frac{\partial \alpha }{\partial x_{1}}\right)}^{2}{\bar{x}}_{2}\right)\\ \; & \leq -\left(k_{1}-\frac{5}{2}\right)∥\bar{z}{∥}^{2}-(k_{2}-\frac{5}{2}∥lQ\Omega^{-1}{∥}^{2})∥\bar{\eta }{∥}^{2}\\ \; & -\left(\varphi_{c}\rho \left(x_{1}\right)-\chi \left(x_{1},k_{m},\lambda \right)\right)x_{1}^{2}-{\bar{x}}_{2}^{2}+(lℵ\left(\nu \right)\\ \; & +\varphi_{c})\rho \left(x_{1}\right)x_{1}^{2} \end{array}$$
where $\chi \left(x_{1},k_{c},\lambda \right)=g+\frac{5}{2}q_{2}+\frac{5}{2}{∥Q\Omega^{-1}E∥}^{2}+\frac{1}{2}{\left(\frac{\dot{k_{m}}}{k_{m}}\right)}^{2}+\frac{1}{2{\left(k_{c}^{2}-x_{1}^{2}\right)}^{2}}+\frac{1}{2}{\left(\frac{k_{c}^{2}-x_{1}^{2}+1}{k_{c}^{2}-x_{1}^{2}}l\right)}^{2}+2{\left(\frac{k_{c}^{2}-x_{1}^{2}+1}{k_{c}^{2}-x_{1}^{2}}\right)}^{2}+2l^{2}$ and $\varphi_{c}$ is a positive constant.Since λ∈Λ is a compact set Λ, it is ensured that there exists $k_{1}$ and $k_{2}$ in (Equation 29) such that $k_{1}\geq 3$ and $k_{2}\geq (\frac{5}{2}∥lQ\Omega^{-1}{∥}^{2}+1)$. There are two positive smoothing functions $\rho (x_{1})$ and $\tau (\lambda ,k_{m})$, which have the property that $\tau \left(\lambda ,k_{m}\right)\rho \left(x_{1}\right)\geq \chi \left(x_{1},k_{m},\lambda \right)+1$. By choosing $\varphi_{c}\geq \tau \left(\lambda ,k_{m}\right)$, we get $\varphi_{c}\rho \left(x_{1}\right)-\chi \left(x_{1},k_{m},\lambda \right)\geq 1$. Consequently, $V_{2}$ is positive definite, and ${\dot{V}}_{2}$ satisfies
(34)$$\dot{V_{2}}\leq -∥\bar{z}{∥}^{2}-{∥\bar{\eta }∥}^{2}-x_{1}^{2}-{\bar{x}}_{2}^{2}+\left(lℵ\left(\nu \right)+\varphi_{c}\right)\dot{\nu },\forall t\geq 0$$For any time ∀t≥0, the simultaneous integration of both sides of (Equation 34) over the interval [0,t) can be obtained as
(35)$$V\left(t\right)-V\left(0\right)\leq \int_{0}^{t}\left(lℵ\left(\nu \left(\tau \right)\right)+\varphi_{c}\right)\dot{\nu }\left(\tau \right)d\tau$$Based on Definition 2 and Lemma 1, for all ∀t≥0, V(t) and ν(τ) are bounded on the interval [0,t). Since $\dot{\nu }=x_{1}^{2}\rho \left(x_{1}\right)$, $x_{1}\left(t\right)$ is square-integrable, and $x_{1}$ and ${\dot{x}}_{1}$ are both bounded, ($x_{1}$ represents the angular error *e*). By invoking Balbarat’s Lemma, we can conclude that $\lim_{t\to \infty }e\left(t\right)=0$. Based on lemma 2, we can deduce that if $|e\left(0\right)|<k_{m}\left(0\right)$, then $|e\left(t\right)|<k_{m}\left(t\right)$, ∀t∈[0,∞). □

## 5. Simulation and Analysis

In this paper, two simulation tasks are carried out based on the micromirror model with the following parameter settings to illustrate the validity of the output performance constraints of the proposed internal-mode backstepping control and the control performance in the presence of anomalous control situations, respectively.

The relevant parameters of the electromagnetic micro-mirror are set as follows: $I_{em}=0.6\times 10^{-11},D_{em}=0.786\times 10^{-9},S_{em}=-0.2828\times 10^{-5}$ and $k_{m}=0.1$. The initial value of the angle of the electromagnetic micromirror is $z_{1}\left(0\right)=0.05$ and D=0.1sin(0.5t).

The micromirror tracking reference signal is $F_{ref}\left(t\right)=\sin\left(5t\right)$, thus the relevant parameters of the reference signal to be tracked are set as follows: $S=\left[\begin{array}{cc} 0 & 5\\ -5 & 0 \end{array}\right]$, $h_{1}\left(0\right)=0$ and $h_{2}\left(0\right)=1$. The parameters of the controller settings are as follows: $\Phi =\left[\begin{array}{cc} -1 & 0\\ 0 & -2 \end{array}\right]$ and $E=\left[\begin{array}{c} 0.1\\ 0.2 \end{array}\right]$.

Firstly, under the premise of ensuring that the micromirror operates normally and is not affected by anomaly control, this investigation sets up the tracking simulation of the reference signal by the PID controller and the internal model backstepping controller proposed in this paper, as well as comparing and analysing the tracking output performance and the error performances of the two kinds of controllers.

Figure 6 and Figure 7 represent the trajectory tracking results and tracking errors of the PID controller without anomaly control. It is evident that the control performance of the PID controller is relatively acceptable only for the middle part of the signal in the transition from the peak to the trough of the sinusoidal signal. However, the error gradually increases as it approaches the peak/valley. Significant overshooting occurs at the peaks/valley, approaching closely the output error threshold preset by the simulation, which is clearly shown in Figure 7.

Figure 8 and Figure 9 denote the trajectory tracking results and tracking errors of the proposed controller without anomaly control. It can be seen that the proposed controller achieves efficient tracking even when relying solely on the error signal of the output angle, which is attributed to the introduction of the extended internal model. The proposed controller does not exhibit obvious overshoot at the peak/valley of the sinusoidal signal. And the control performance exhibited is consistent with the performance in the intermediate signal portion of the peak-to-trough transition, which is due to the constraining effect of the BLF on the output performance.

Figure 10 shows a comparison of the control performance of the two controllers on the sinusoidal signal, further illustrating that the control output performance of the proposed controller based on output regulation is superior overall to the control performance of the PID controller that uses more state information.

Secondly, by introducing the case of anomaly control (control signal reversal) in micromirror control simulation, this investigation further explores the tracking performance of the PID controller and the proposed controller on the reference signal under abnormal conditions. Through the comparative analysis, the output performance and error control capability of the two controllers in the face of the anomalous control situation are evaluated.

Figure 11 and Figure 12 indicate the trajectory tracking results and tracking errors of the PID controller when anomaly control occurs. It is observed that the control performance of the PID controller is significantly affected when anomaly control occurs, which indicates the lack of robustness of the PID control in anomaly control situations. Figure 12 demonstrates that the resulting error when anomaly control occurs exceeds the output error threshold preset by the simulation. Additionally, comparing Figure 12 with Figure 7, the steady state output performance after the occurrence of anomaly control is degraded due to the fact that the PID control lacks a mechanism to effectively handle anomaly control situations.

Figure 13 and Figure 14 depict the trajectory tracking results and tracking errors of the proposed controller when anomaly control occurs. Although the control performance of the proposed controller shows some deviation under anomaly control conditions, the tracking error is effectively controlled within the preset threshold because of the BLF. Comparing Figure 14 with Figure 9, it is obtained that the steady state control performance of the proposed controller does not present a significant degradation similar to the PID controller after the occurrence of anomaly control.

Figure 15 showcases the comparison of the tracking error of the two controllers in case of anomaly control, and the results demonstrate that the output performance of the proposed controller outperforms the output performance of the PID controller.

## 6. Conclusions

In this paper, the problem of asymptotic tracking of the micromirror model with output performance constraints and anomaly control is addressed, and an internal model backstepping control strategy based on output regulation is proposed. Simulation results show that the output performance of the proposed controller at the peaks and valleys of the sinusoidal signal improves by 85% compared with the PID controller. In addition, the controller can effectively control the output error within a predefined error threshold under anomalous control situations, displaying excellent robustness.

For optical devices developed based on electromagnetic micromirrors, e.g., LIDAR, and depth cameras, the proposed controller contributes to accuracy and durability, and enhances reliability under anomalous control. Future work will consider incorporating sliding mode control to reduce the effects of anomaly control, allowing the micromirror system to reach steady state control more quickly after recovering from anomaly control to normal control.

## Figures and Tables

**Figure 1 micromachines-15-00925-f001:**
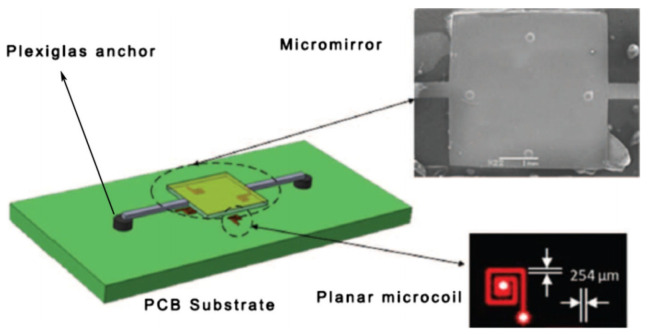
The entire structure of the micromirror.

**Figure 2 micromachines-15-00925-f002:**
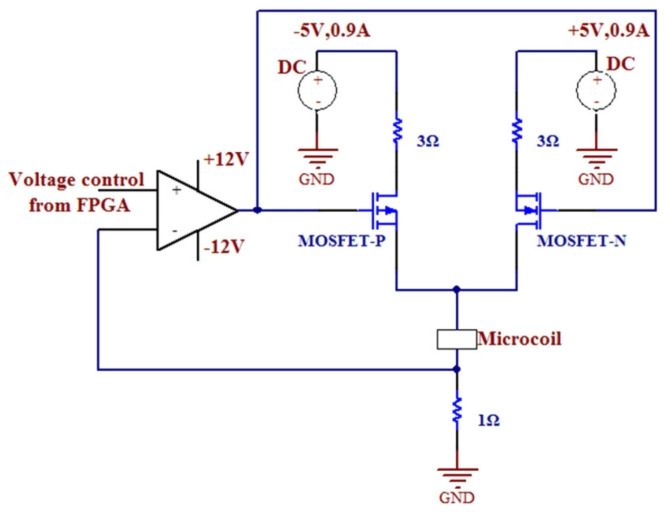
The structure of the VCCA circuit for the electromagnetic micromirror.

**Figure 3 micromachines-15-00925-f003:**
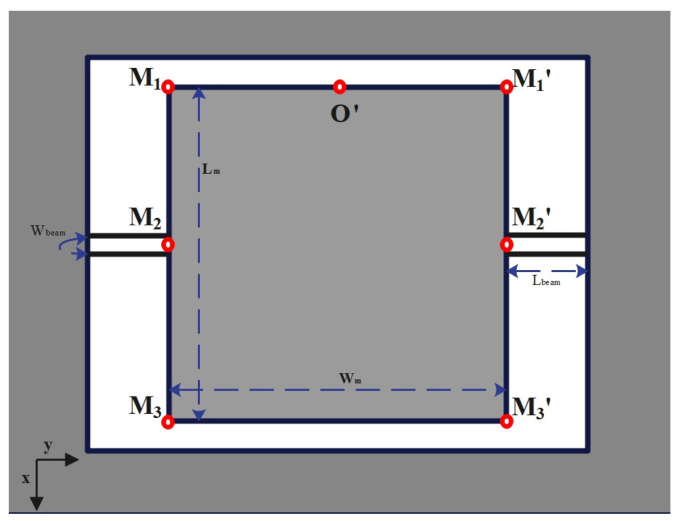
The overhead view of the micromirror.

**Figure 4 micromachines-15-00925-f004:**
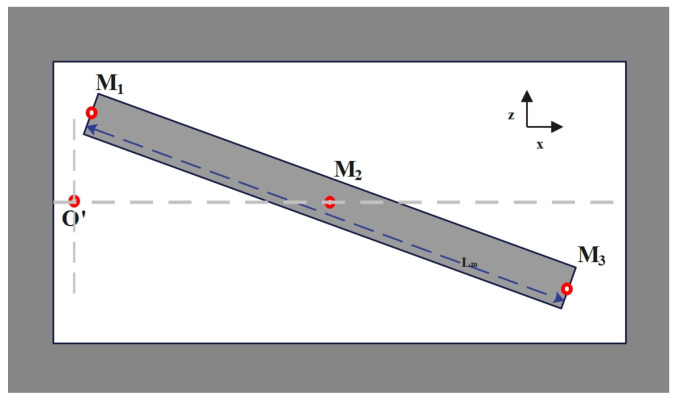
The lefthead view of the micromirror.

**Figure 5 micromachines-15-00925-f005:**
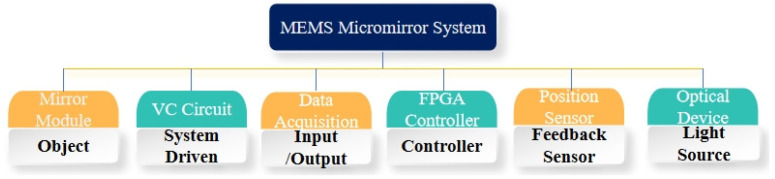
The composition of the micromirror system.

**Figure 6 micromachines-15-00925-f006:**
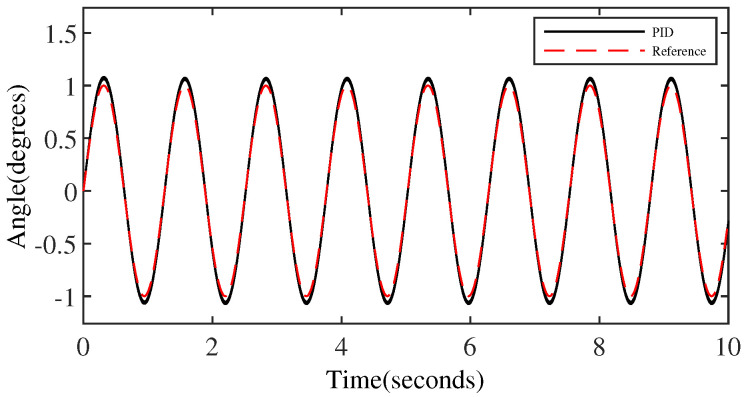
The output performance of the PID controller.

**Figure 7 micromachines-15-00925-f007:**
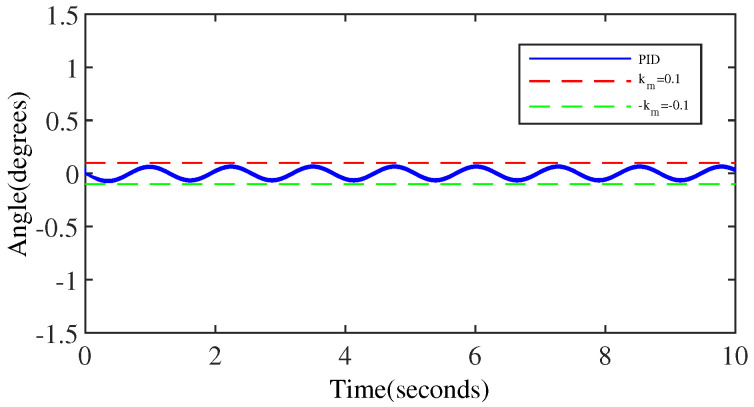
The tracking error of the PID controller.

**Figure 8 micromachines-15-00925-f008:**
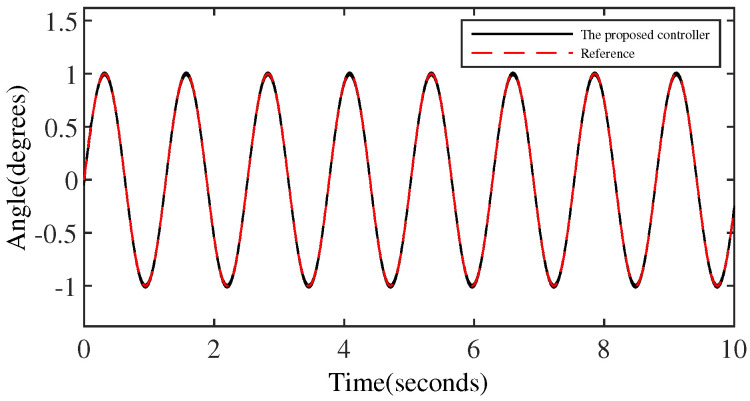
The output performance of the proposed controller.

**Figure 9 micromachines-15-00925-f009:**
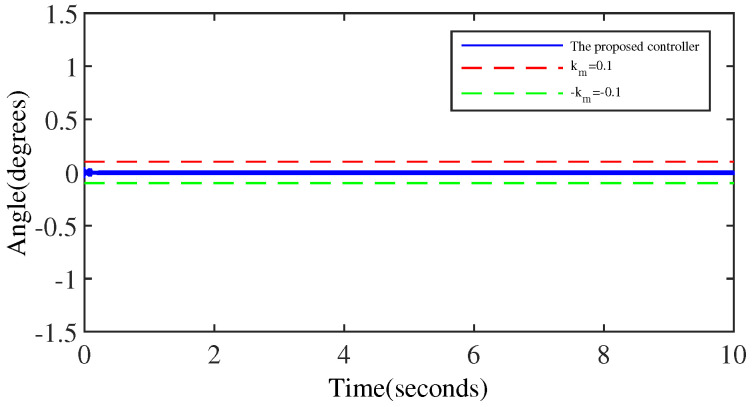
The tracking error of the proposed controller.

**Figure 10 micromachines-15-00925-f010:**
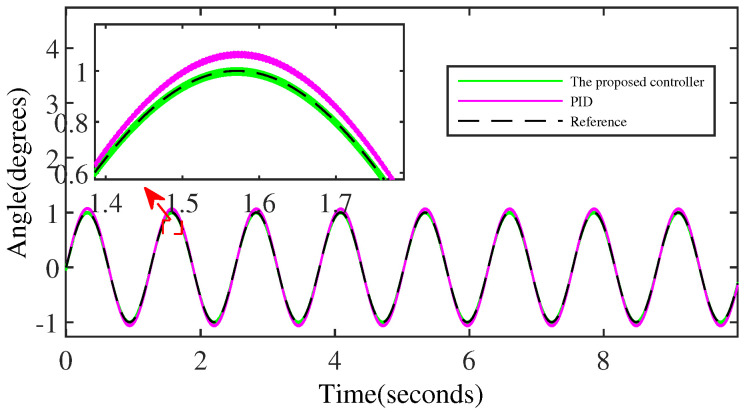
The comparison of the output performance of the two controllers.

**Figure 11 micromachines-15-00925-f011:**
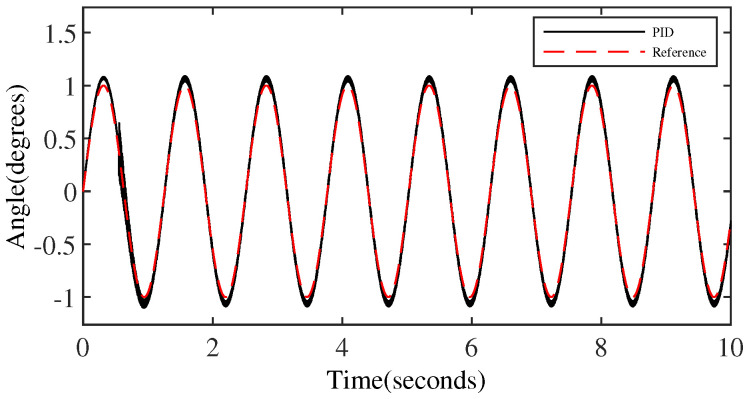
The output performance of the PID controller (anomaly control exists).

**Figure 12 micromachines-15-00925-f012:**
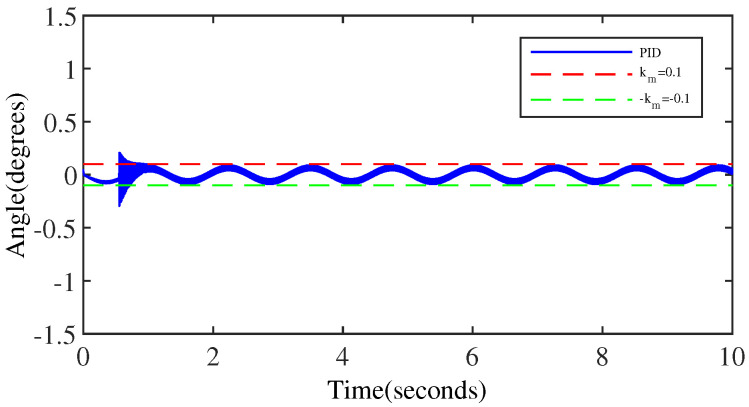
The tracking error of the PID controller (anomaly control exists).

**Figure 13 micromachines-15-00925-f013:**
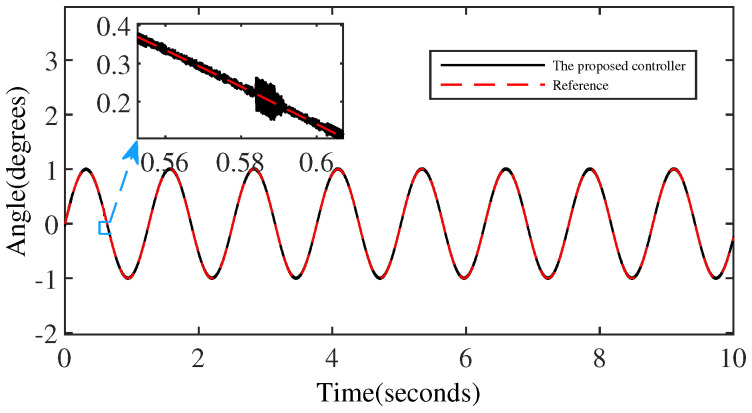
The output performance of the proposed controller (anomaly control exists).

**Figure 14 micromachines-15-00925-f014:**
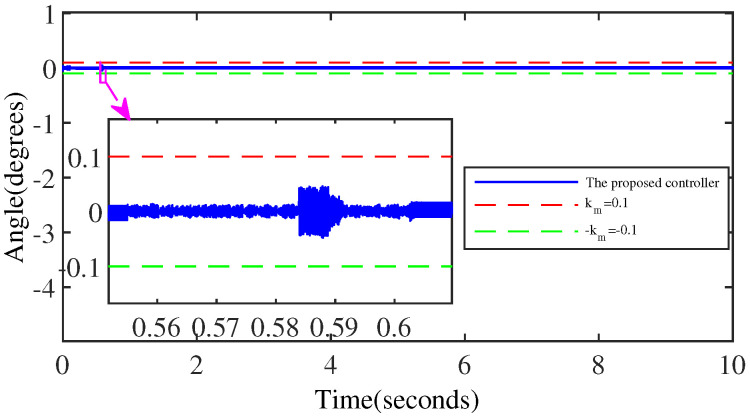
The tracking error of the proposed controller (anomaly control exists).

**Figure 15 micromachines-15-00925-f015:**
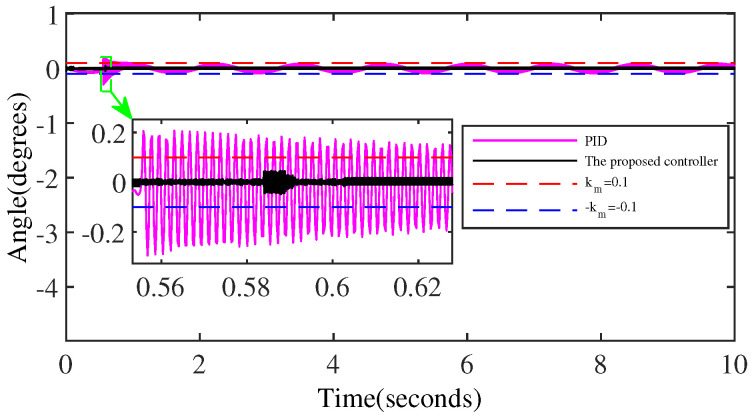
The comparison of the tracking errors of the two controllers in anomaly control situations.

## Data Availability

The original contributions presented in the study are included in the article, further inquiries can be directed to the corresponding author.

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
