# Peer review of "Adaptive Internal Model Backstepping Control for a Class of Second-Order Electromagnetic Micromirror with Output Performance Constraints and Anomaly Control"

_micromachines, 2024, doi:10.3390/mi15070925_

Round 1

Reviewer 1 Report

Comments and Suggestions for Authors

Dear authors,

The article investigated the asymptotic tracking problem for a class of second-order electro-magnetic micromirror model with output performance constraints and anomaly control. There are some details that the author needs to further clarify.

Please find the attached file for details.

Reviewer 2 Report

Comments and Suggestions for Authors

This article proposes an adaptive output feedback control scheme based on an extended internal model. The developed controller takes into account uncertainties in system parameters and external disturbances when working with input data. This article takes a comprehensive look at the impact of performance limitations and anomaly control on the micromirror system, it can ensure the output performance of the micromirror system even if an anomaly occurs during the control process, avoiding collisions between the micromirror and the flat drive coil due to excessive deflection angle.

The article as a whole is relevant, the task posed is interesting for research. The presented results have scientific novelty.

There are minor requests for changes to the article.

- From my point of view, the Introduction is long, it is recommended to split it into the introduction itself and a literature review or background

- The last paragraph of the introduction should contain descriptions of the remaining sections, this is necessary for navigating through the article

- What is Sh in equations (9) and (10)

- in equation (10) the function J (.) is the same J (/lambda) above on line 243

- in line 273 the error is "assumption 3" instead of "assumption 2"

- the “R-type Nussbaum function” introduced in Definition 1 requires clarification and clarification, the definition is very general

- on line 352 "Lyapunov stability" requires explanation, are Lyapunov functions used?

- In conclusion, you can add prospects for the development of research or completeness for application in conditions of limitations
